# Donor-Derived Cell-Free DNA (ddcf-DNA) and Acute Antibody-Mediated Rejection in Kidney Transplantation

**DOI:** 10.3390/medicina57050436

**Published:** 2021-05-01

**Authors:** Vishal Jaikaransingh, Pradeep V. Kadambi

**Affiliations:** Department of Medicine, Division of Nephrology, University of Florida, 655 West 8th Street, C290, Jacksonville, FL 32209, USA; pradeep.kadambi@jax.ufl.edu

**Keywords:** donor-derived cell-free DNA, cell-free DNA, kidney transplantation, antibody-mediated rejection

## Abstract

Monitoring kidney transplant recipients for evidence of allograft rejection is essential to lower the risk of graft loss. The traditional method relies on serial checks in serum creatinine with a biopsy of the allograft if dysfunction is suspected. This is invasive, labor-intensive and costly. As such, there is widespread interest in the use of biomarkers to provide a noninvasive approach to detecting allograft rejection. One such biomarker is donor-derived cell-free DNA (ddcf-DNA). Here, we review the methodology for the determination of the amount/fraction of ddcf-DNA, evaluate the available data of its use in kidney transplantation and render an opinion in the clinical decision-making of these patients.

## 1. Introduction

Early studies involving cell-free DNA (cf-DNA) were performed in the 1970s in patients with cancer [1]. In the field of solid organ transplantation, donor-derived cell-free DNA (ddcf-DNA) was first reported in the 1990s [2,3]. In two independent publications, Lo et al. [2] and Zhang et al. [3] described the presence of ddcf-DNA by the detection of Y chromosomal DNA of male donors in the serum and urine, respectively, of female recipients. Since then, advances in the field have allowed for detection using commercially available assays [4,5,6]. The amount of donor-derived cell-free DNA is usually expressed as a percentage of the total cell-free DNA present. These assays do not require genotyping of the donor or the recipient [7]. The technology involves the detection of disparate single nucleotide polymorphisms (SNPs) across the whole genome and allows for the separation of the DNA derived from two individuals, and is known as genome transplant dynamics (GTD). Elevated levels of ddcf-DNA point to injury of the allograft and in some patients can suggest acute rejection.

## 2. Determination of ddcf-DNA

The methods available for the detection and quantification of ddcf-DNA in recipient plasma utilize the polymerase chain reaction (PCR). The options include real-time PCR, droplet digital PCR and massive parallel sequencing, also known as next-generation sequencing (NGS) [8,9]. Digital PCR and NGS do not require donor genotyping and they assess for a single nucleotide polymorphism for which the recipient is homozygous for a particular allele but the donor is not [7]. Both methodologies have been validated clinically and are utilizing commercially available assays.

At this time, there is no standardization for targeted SNPs in commercially available assays and the targets vary. Two NGS assays, AlloSure (Care Dx, Inc., Brisbane, CA, USA) [4,5] and Prosepera (Natera, Inc., San Carlos, CA, USA) [6] target 266 SNPs and 1392 SNPs, respectively. Melancon et al. compare the performance of these assays in a small single-center study involving 76 transplant recipients [10]. The study participants had a paired blood draw with one sample being allocated to each assay. The results of the each assay were correlated with the biopsy of the allograft scored using the Banff 2017 classification. Both assays were not statistically different with respect to specificity, sensitivity and positive and negative predictive values. Allosure, however, demonstrated a shorter turnaround time for 75% of patients from the blood draw to a result; and when compared with that of Prosepera, the difference was at least one day.

## 3. The Kinetics of ddcf-DNA after Kidney Transplantation

The vast majority of cf-DNA arises from hematopoietic cells undergoing apoptosis [11,12]. Only a small amount comes from solid organs [13]. It appears to be cleared rapidly from plasma with a reported half-life ranging from 16 min to 13 h in studies looking at the rate of disappearance of fetal cell-free DNA from maternal plasma [14,15].

Following kidney transplantation, ddcf-DNA decreases rapidly in the recipient plasma and this is reflected in a decline in the percentage of cf-DNA, which is donor-derived as compared to recipient-derived. Gielis et al. [16] evaluated the change in plasma ddcf-DNA in 42 stable kidney transplant recipients using NGS in a cohort which included both living and deceased donors (7 living donors and 35 deceased donors). They demonstrated an exponential decline from a median of 10.2% of ddcf-DNA (range 2.6% to 41.9%) on the first postoperative day, with a mean of 0.46% (+/− 0.21%) being reached on postoperative day 9.85 (+/− 5.6 days). In this cohort, the recipients of the living donor organs had lower levels of cf-DNA on postoperative day 1 when compared to the recipients of the deceased donor organs, but both groups declined to similar levels by postoperative day (POD) 10.

Shen et al. [17] conducted a similar study but in addition to ‘stable’ kidney transplant patients, patients with delayed graft function (DGF) were also included. Their cohort consisted of 7 living donor and 14 deceased donor recipients. In the deceased donor arm, 6 patients had DGF. The ddcf-DNA fraction declined rapidly from a median of 20.69% at 3 h after reperfusion to 5.22% on POD 1 (16.4 h post-transplant) to 1.98% on POD 2 and 0.85% on POD 7. The concentration of ddcf-DNA was significantly lower on POD 2 compared to POD 1 (*p* = 0.039) but for all the other days there was not a statistically significant difference when compared to the prior day. The concentration of ddcf-DNA was significantly higher for patients receiving transplants from deceased donors when compared to living donors, at 44.99% compared to 10.24% at 3 h post reperfusion (*p* < 0.01). In addition, the rate of decline in the ddcf-DNA fraction was also slower for the recipients of the deceased donor organs. The higher fraction of ddcf-DNA persisted in the deceased donor recipients at POD 7 at 1.1% compared to 0.59% for the recipients of the living donor organs (*p* < 0.05). In addition, there was a higher concentration of ddcf-DNA in the patients who had DGF compared to those who did not, but this did not reach statistical significance.

After stabilization of the ddcf-DNA fraction after transplant, the levels can rise as a result of acute rejection but can also rise when the allograft suffers other types of injury including acute tubular necrosis [18], and in the cases of severe post-transplant infection including allograft pyelonephritis and BK virus nephropathy (BKVN) [4,18,19].

## 4. Ddcf-DNA and Acute Antibody Mediated Rejection

Despite considerable interest in ddcf-DNA as a convenient, noninvasive means of diagnosing acute rejection, published data has for the most part been restricted to small cohorts with inconsistent methodologies. This is highlighted in this systematic review of the studies published through June 2018 by Knight et al. [20]. It included 739 kidney transplant recipients with 509 from papers published only in abstract form. Moreover, multiple techniques for measuring ddcf-DNA were reported. However, as commercial assays have become available there has been more consistency across publishing literature.

The AlloSure assay has been utilized in several published studies [4,21]. The “Multicenter Circulating Donor-Derived Cell Free DNA in Blood for Diagnosing Acute Rejection in Kidney Transplant Recipients” (DART) study by Bloom et al. prospectively examined 107 biopsies in 102 patients, 6 of which had active AMR4. Utilizing a 1% cut off, there was a sensitivity of 59% and specificity of 85% with a PPV of 61% and NPV of 84% for any form of acute rejection. The median ddcf-DNA was 2.9% for AMR compared to 1.6% for any form of acute rejection (1.2% for Banff 1B TCMR and 0.2% for Banff 1A TCMR). This gave an AUC of 0.84% with 81% sensitivity and 83% specificity for AMR, and PPV and NPV of 44% and 96%, respectively.

Huang et al. performed a retrospective analysis of 63 patients from a single center who had an assessment of ddcf-DNA using Allosure within 30 days of allograft biopsy [21]. The biopsies were performed “for cause” basis to assess for rejection because of graft dysfunction for the development of de novo DSA. Acute rejection was diagnosed in 34 patients of which 24 were AMR (2 of these were mixed AMR and TCMR). The median ddcf-DNA fraction was higher in patients with antibody-mediated rejection at 1.35%, compared to 0.27% with isolated T cell-mediated rejection and 0.38% with no rejection. Of note, 28% of the patients who did not have rejection (8 of 29) had a ddcf-DNA fraction greater than 1%, with one patient having a fraction as high as 5.2%. Using a threshold of 0.74%, the sensitivity for AMR was 100% with a negative predictive value of 100%. However, using such a low threshold resulted in a specificity of 71.8% and a positive predictive value of 68.6%.

Another single-center study by Zhang et al. prospectively compared 18 patients with AMR to 19 patients with stable allograft function [22]. Here, an NGS assay targeting 56,049 SNPs was used. The median ddcf-DNA fraction was 2.4% in the group with antibody-mediated rejection compared to 0.65% in the group with stable allograft function (*p* < 0.001). Using a cut off of 1%, the sensitivity was 88.9%, specificity was 73.7%, PPV was 76.2% and NPV was 87.5%. The patients without AMR but with DSA in the stable group had a median ddcf-DNA fraction of 1.09% and when this group was compared to patients with AMR, the results were not statistically significant (*p* = 0.074).

There are limited data looking at ddcf-DNA in repeat kidney transplant recipients (RKTRs). Mehta et al. examined a cohort of 12 RKTRs (11 with two allografts and 1 with three allografts in situ) from the DART study, and compared the plasma levels of ddcf-DNA in these patients to a cohort of 202 single kidney transplant recipients (SKTRs) from the same study [23]. There was no documented rejection in either of these cohorts and ddcf-DNA was checked as part of surveillance. It was found that serum ddcf-DNA levels were significantly higher in the RKTRs group when compared to the SKTRs group (0.29% vs. 0.19%, respectively, *p* < 0.001). In the same study, the authors also evaluated serum ddcf-DNA levels in 11 RKTRs (9 patients with two, 1 patient with three and 1 patient with four kidney allografts in situ) from the DART study who had a kidney biopsy of their most recently transplanted allograft for a clinical indication. Within this cohort, 6 patients were diagnosed with rejection (2 TCMR and 4 AMR) and 5 had no rejection findings in the biopsy. The ddcf-DNA in RKTRs with rejection was higher than in RKTRs without rejection (median 1.36% vs. 0.41%, respectively, *p* = 0.009). Of note, the AlloSure method that was used in this study to determine ddcf-DNA levels does not require genotyping of the donor(s). As such, it does not distinguish which allograft(s) may have contributed to the total cf-DNA measured. Hence, ongoing inflammation in prior allografts could be a confounder resulting in higher levels of cf-DNA at the baseline in patients with RKTRs.

Pediatric kidney transplant patients are a special population where avoidance of a kidney biopsy in favor of a less invasive approach may be desired because of the potential distress. Unfortunately, data for this patient population are also extremely limited. Puliyanda et al. examined 67 pediatric kidney transplant recipients who underwent transplantation at two centers between October 2017 and October 2019 [24]. During that time, the patients underwent an assessment of ddcf-DNA using the AlloSure assay, either as part of routine monitoring (19 patients) or in response to a clinical suspicion of rejection (48 patients). In the cohort where there was clinical suspicion of rejection, 21 (43.8%) patients had a ddcf-DNA of >1% and all underwent biopsy. All the biopsies showed evidence of rejection (22.9% AMR, 4.2% TCMR and 16.7% mixed AMR and TCMR). In addition, 7 patients underwent a biopsy with a ddcf-DNA of <1%. Of these, 3 patients showed evidence of rejection (1 each of AMR, TCMR and mixed AMR and TCMR) and 4 had no rejection in the biopsy. The ddcf-DNA fraction of >1% had a sensitivity of 86% and a specificity of 100% (*p* = 0.002) for the diagnosis of acute rejection.

Collectively, these studies indicate that with rejection, particularly AMR, the ddcf-DNA fraction in plasma rises. This elevation, however, is not exclusive to rejection. Gielis et al. used an NGS assay targeting 1027 SNPs in 107 patients with 792 longitudinally collected blood samples. Their cut off was set at a ddcf-DNA fraction of 0.88% and 13% of the samples had fractions above the cut off. The increases in the ddcf-DNA fraction were significantly associated with acute rejection (*p* = 0.017), AT (*p* = 0.011) and acute pyelonephritis (*p* = 0.032). The AUC for an acute rejection of 0.64 was no different than that of serum creatinine. The sensitivity and specificity for ddcf-DNA were 38% and 85%, respectively.

Table 1 summarizes the studies assessing ddcf-DNA in plasma for the diagnosis of acute rejection using commercially available assays.

## 5. Discussion

The majority of published studies demonstrate elevated levels of ddcf-DNA in acute rejection. This association seems to be strong for AMR when compared to TCMR. However, there are multiple other causes of elevated serum ddcf-DNA including infection and acute tubular necrosis. Both of these are common occurrences in a kidney transplant population. The comprehensive evaluation of published literature is confounded by small sample sizes, heterogeneity in methodologies and differences in proposed diagnostic thresholds. However, there seems to be more consistency in recently published data because of the availability of commercial assays. These assays appear to be roughly equivalent but were only compared in one study.

There has not been a randomized trial comparing the performance of donor-derived cell-free DNA to the current standard of care, and although the data outlined suggests a high NPV for ddcf-DNA in antibody-mediated rejection, studies were for the most part small and/or single center. In addition, it is unclear how these assays will perform in a group of high-risk patients, such as those with high pre-transplant titers of donor-specific antibody or in the development of de novo donor-specific antibodies after transplantation.

At this time, there is insufficient evidence to recommend using ddcf-DNA in isolation in routine clinical practice. The gold standard for diagnosis of antibody-mediated rejection remains to be biopsy of the allograft. If a clinical indication for biopsy exists, the level of ddcf-DNA below the defined threshold should not preclude biopsy. For now, ddcf-DNA measurements may still find a role clinically at times when biopsies are contraindicated. There have also been suggestions that regular monitoring of ddcf-DNA may help to detect subclinical rejection, guide an individualized approach to immunosuppression or to follow the response to treatment of acute rejection. At this time there is insufficient evidence for these tests to be applied routinely.

## 6. Conclusions

At its core, ddcf-DNA is an indicator of allograft injury. It is not specific for any form of rejection including antibody-mediated rejection. Elevated levels can occur in processes limited to the allograft, such as rejection and ATN but can also be elevated in systemic conditions such as malignancy and infection. There are still many unanswered questions such as the ideal methodology for determination of ddcf-DNA and the optimal threshold for the diagnosis of rejection. More data are needed, particularly randomized trials comparing ddcf-DNA to the current standard of care. At this time, no clear recommendations can be made and kidney allograft biopsy remains the gold standard for the diagnosis of antibody-mediated rejection.

## Figures and Tables

**Table 1 medicina-57-00436-t001:** Studies assessing ddcf-DNA in plasma for the diagnosis of acute rejection using commercially available assays.

Study	Methodology	Number of Patients	Threshold (% dd-cfDNA)	Sen/Sepc	PPV/NPV
Bloom et al. [4]	NGS (AlloSure)	102P/107B	1%	59/85–Any Rejection 81/83-AMR	61/84–Any Rejection 44/96-AMR
Sigdel et al. [6]	NGS (Prospera)	217B	1%	88.7/72.6–Any Rejection	52/95–Any Rejection
Huang et al. [21]	NGS (AlloSure)	63P	0.74%	79/72–Any Rejection 100/71.8-AMR	77/75–Any Rejection 68.6/100-AMR
Mehta et al. [23]	NGS (AlloSure)	11P/11B	1%	NR	NR
Puliyanda et al. [24]	NGS (AlloSure)	67P/28B	1%	86/100–Any Rejection	NR

B—number of biopsies; NR—not reported; P—number of patients.

## Data Availability

The data presented in this study are openly available (PubMed).

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
