# Peer review of "Donor-Derived Cell-Free DNA (ddcf-DNA) and Acute Antibody-Mediated Rejection in Kidney Transplantation"

_medicina, 2021, doi:10.3390/medicina57050436_

Round 1

Reviewer 1 Report

The authors have presented an evidence based practice for Cell-Free DNA and rejection in kidney transplantation. However they need to make following changes to the paper. 

1) Add references and literature search on role of Cell-free DNA in re-transplants

2) Add evidence of role of cell-free DNA in pediatric transplant patients. (Emerging new evidence.)

3) Best for authors to draw a table to show the studies performed, patient numbers, type of studies, Cut off values for Cell-Free DNA and their relation to diagnosis of rejection. 

Changes the paper:

Line 

11: Remove rapid, accurate. Cell-free DNA is not rapid and not accurate. 

15: Remove 'as to How to incorporate it'

20:  Remove 'and involved its detection'

136: Edit - 'more consistency in recently published '. More was used twice- does not read well. 

146 and 147. Re-phrase the lines

148-149: Remove ' for the test'

153: Edit ' for these tests to be applied routinely'

155: Edit:  'is an indicator of allograft injury'   

Author Response

Dear Reviewer,

We want to thank you for your feedback and we have now added:

  1. A section on the role of Cell-free DNA in re-transplants
  2. A section on pediatric kidney transplants
  3. A table summarizing the studies

We have also incorporated all the other changes you have suggested in the body of the manuscript.

Please accept our thank you in helping us making this a much better manuscript.

Reviewer 2 Report

In this paper the authors review the literature if the Donor-derived cell-free DNA could be used for diagnosing Acute Antibody mediated rejection.

Donor-derived cell-free DNA test have been available since the 1990s, but their usefulness in clinical practice has not been established.

The authors give a summary of the clinical problem and background information on the available assays, presenting both the advantages and disadvantages.

They refer to several clinical trials, presenting the major findings and conclusions. This selection of trials gives the reader a good overview of this topic.

The authors discuss the limitations of these assays and conclude that biopsies cannot be avoided.

The paper is nicely written, logical, easy to read. The English language and style is excellent. It covers the important trials in this field and summarises the results. 

Author Response

Dear reviewer,

We want to thank you for your kind comments and the valuable feedback you have provided us!

Round 2

Reviewer 1 Report

Thank you very much to the authors for making all the recommended changes.

In the same study, the office is also. Line 126

Please correct ''the office''. This is wrong use of words.  

Author Response

Dear Reviewer,

Thanks so much for your kind words. We apologize that we uploaded the incorrect version for your review. This is the corrected version. What you have mentioned in line 126 should have read "the authors also" instead of "the office is also". We've also rectified a few minor issues. We hope that you will be OK with this change. Again, we thank you for your review to make this manuscript significantly better.

The Authors